# Association between Dietary Patterns and Handgrip Strength: Analysis of the Korean National Health and Nutrition Examination Survey Data Between 2014 and 2017

**DOI:** 10.3390/nu12103048

**Published:** 2020-10-05

**Authors:** Yunkoo Kang, Jieun Kim, Do-Yeon Kim, Seung Kim, Sowon Park, Hyunjung Lim, Hong Koh

**Affiliations:** 1Department of Pediatrics, Yonsei University Wonju College of Medicine, Wonju 26426, Korea; monkeydluffy@yonsei.ac.kr; 2Department of Pediatrics, Yonsei University College of Medicine, Severance Children’s Hospital, Seoul 03722, Korea; pedks@yuhs.ac (S.K.); sowon81@yuhs.ac (S.P.); 3Research Institute of Medical Nutrition, Kyung Hee University, Seoul 02447, Korea; jieunkim@khu.ac.kr (J.K.); rou_@naver.com (D.-Y.K.); 4Department of Medical Nutrition, Graduate School of East-West Medical Science, Kyung Hee University, Yongin 17104, Korea

**Keywords:** diet, handgrip strength, nutrition, dietary pattern, children and adolescents, KNHANES

## Abstract

Non-invasive anthropometric measurement methods such as those for measuring height and weight are crucial in pediatric patients. However, research focusing on the association between the type of dietary pattern and handgrip strength and handgrip-to-weight ratio in adolescents has not been carried out yet. This cross-sectional analysis of the 2014–2017 Korean National Health and Nutrition Examination Survey assessed 2327 adolescents (aged 10–18 years) who had their handgrip strength measured and analyzed its association with dietary pattern. The clusters were examined for nutritional values, and the *ready-to-eat*, *balanced,* and *Western-style fast-food* clusters were ultimately generated. Overall, 85.6% of the participants were assigned to a *ready-to-eat* dietary pattern, 9.3% to a *Western-style fast-food* dietary pattern, and 5.1% to a *balanced* dietary pattern. Compared with the participants following a *balanced* dietary pattern, those following a *ready-to-eat* dietary pattern were shown to have a significantly lower handgrip strength and handgrip-to-weight ratio. Decreased handgrip strength and handgrip-to-weight ratio values in participants following *ready-to-eat* dietary patterns indicate a diffuse problem in adolescents’ health and possibly imply an association between reduced muscle quality and dietary pattern. Therefore, the overall environmental factors potentially inducing such unhealthy dietary preferences should be investigated, and appropriate lifestyle changes in Korean adolescents should be encouraged.

## 1. Introduction

Non-invasive anthropometric measurements methods, such as those for measuring height and weight, are crucial for pediatric patients [1,2,3]. In fact, anthropometric measurements can help the decision-making process about the need for additional tests. In this context, muscle strength has recently been recognized as an essential measurement [4,5,6]. Although muscle strength is a crucial factor for muscular force exertion, several other factors are known to be involved. An individual with high muscle strength is likely to follow an appropriate diet as well as a regular physical activity schedule, in addition to being in a healthy psychological state. In fact, several studies have reported an association between sarcopenia and obesity, metabolic syndrome, non-alcoholic fatty liver disease (NAFLD), and even mental disorders [7,8,9]. Skeletal-muscle mass can be assessed by either dual-energy X-ray absorptiometry or bioelectrical impedance analysis and magnetic resonance imaging (MRI); ultrasound can be used to measure skeletal muscle quantity [10]. Furthermore, muscle strength can be accessed by plank exercise, modified pull-up, knee extension exercises, and handgrip strength. Nevertheless, measuring handgrip strength (HGS) is a simple and inexpensive way to evaluate muscle strength [11,12,13]. Furthermore, muscle quality can be estimated by dividing muscle strength by muscle mass size. However, in a simple way, we can predict relative muscle strength by dividing handgrip strength by body weight. Moreover, HGS has been shown to be associated with many diseases in children, such as metabolic syndrome and NAFLD [11].

Adequate dietary intake is essential for the health of children and adolescents. Unbalanced dietary-intake habits are known to contribute to obesity in adolescents [14,15]; additionally, they are associated with major health problems, including metabolic syndrome, mental disorders, and even cancer [16]. Moreover, chronic conditions such as adolescent obesity are likely to continue and even worsen in young adults [17]. According to several studies, balanced dietary intake is essential for both adolescents and adults, since it can affect the whole life span.

Infectious diseases, mental disorders, alcohol and/or drug abuse, nutrition deficiencies, obesity, and physical inactivity are the primary health risks for adolescents [18]. Moreover, nutritional inadequacies (e.g., undernutrition, overnutrition, picky eating) are known for being common causes of several pathologies in adolescents [19]. On the other hand, many studies have demonstrated a decrease in handgrip strength in the pediatric population [5,8,12,13]. However, the association between dietary pattern and handgrip strength and handgrip-strength-to-weight ratio in adolescents has not been thoroughly investigated yet. In this study, we investigated the association between handgrip strength and dietary intake patterns in Korean adolescents. 

## 2. Materials and Methods

### 2.1. Study Population

We collected data from the 2014–2017 Korean National Health and Nutrition Examination Survey (KNHANES), a nationally representative cross-sectional survey carried out by the Korea Center for Disease Control and Prevention (KCDC) every 3–4 years. The scope of KNHANES is to assess the health status of the Korean population by using a multistage, clustered, stratified, and rolling sampling method [20]. This study was conducted in accordance with the guidelines of the Declaration of Helsinki (1964), and its protocol was approved by the Institutional Review Board (IRB) of the KCDC. Informed consent was obtained from all participants at the time of the survey. The Kyung Hee Institutional Review Board approved this study protocol (KHSIRB-20-052(EA)).

We initially collected data on a total of 2988 subjects from the 2014–2017 KNHANES database; thereafter, we excluded non-adolescent subjects (i.e., ≥19 years old or <10 years old) (*n* = 340), as well as subjects with extreme energy-intake levels (i.e., <500 kcal/day or >5000 kcal/day) (*n* = 41) and subjects without height, weight, and/or HGS data (*n* = 280). In total, 2327 participants were eventually included in our study (Figure 1).

### 2.2. Assessment of Socioeconomic Data and Physical Activity 

Sociodemographic data of the study participants included age, house income level, province, and physical activity level. Physical activity was defined using the metabolic equivalent task (MET) score according to the scoring protocol of the Korean Physical Activity Questionnaire short form [21]. Physical activity was categorized according to the total MET score as inactive (<600 MET-min/week), active (600–3000 MET-min/week), or health-enhancing (>3000 MET-min/week).

### 2.3. Anthropometric Measurements

Height (Seca 225; SECA, Hamburg, Germany), weight (GL-6000-20; CASKOREA Co., Ltd., Seoul, Korea), and waist circumference (Seca 200; SECA, Hamburg, Germany) were measured by trained medical staff using standardized techniques and calibrated equipment. Body mass index (BMI) was calculated as weight (kg) divided by height squared (m^2^). The 2017 Korean Children and Adolescents Growth Standard values were used for anthropometric assessment and BMI age- and sex-specific reference data [22]. Blood pressure (BP) was measured before and after a 5-min interval, and the mean value of the two measurements was implemented in our analysis.

### 2.4. Measurement of Handgrip Strength

HGS was measured thrice for each hand using a digital grip strength dynamometer (TKK 5401; Takei Scientific Instruments Co., Ltd., Tokyo, Japan). All participants were instructed to hold the dynamometer in an upright standing position with the arms by their sides and to squeeze the grip with full force for 3 s each time. The mean of three trials was used for data analysis. Handgrip-strength-to-weight ratio (HGSWR) was determined by the following equation: HGS/body weight (kg) × 100 [11,12,13].

### 2.5. Dietary Intake Assessment and Food Grouping

One week after the health interview and examination, dietary intake was assessed using the single 24-h diet recall method through home interviews by trained interviewers. Nutrient intake was calculated by using the Korean Foods and Nutrients Database of the Rural Development Administration [23,24].

Food data were categorized into 25 food groups by using the Korean Nutrient Database. Food types were divided into 18 groups (grains, potatoes, sugars, beans, nuts, vegetables, mushrooms, fruits, meats, eggs, fish, seaweeds, dairy products, oils, beverages, seasonings, processed foods, and other) in the Korean Nutrient Database to simplify the interpretation of components. Due to the high intake rate of grains and their derivatives in Korea, we further divided this food group into four subgroups based on their nutritional profiles: white rice, other grains, wheat and bread, and noodles. Additionally, kimchi—a traditional fermented-cabbage-based food—was also separated from the vegetable group, since it is a commonly consumed side dish in Korea. Ultimately, the beverages group was further divided into soda, sugar-sweetened beverages (SSBs), and non-sugar beverages, based on sugar content; in addition, alcohol was removed from the group due to the age of the participants.

### 2.6. Dietary Pattern Analysis

Dietary patterns were generated by a k-means cluster analysis using SAS FASTCLUS based on the gram per day (g/day) percentage of total energy contribution for each food group. This k-means procedure generates clusters by comparing the Euclidean distance between each subject. Two to five solutions were examined to assess which cluster set was more suitable for the determination of the dietary patterns. The content of the clusters was examined for its nutritional value, and a three-cluster set was eventually selected. Three types of dietary patterns were thereby identified and named according to the food groups with high consumption: the *ready-to-eat* dietary pattern, which contained only white rice, eggs, fish, dairy products, and processed food; the *Western-style fast-food* dietary pattern, which was based on wheat/bread, noodles, sugars, meat, oils, sodas (sweetened and unsweetened), and seasonings; and the *balanced* dietary pattern, which was based on whole grains, potatoes, beans, nuts, vegetables, mushrooms, fruits, eggs, fish, seaweed, and dairy products.

### 2.7. Statistical Analysis

Statistical analysis was performed using PROC SURVEY in SAS to take into account the complex sampling design and appropriate sampling weights for the national survey. The k-means procedure was used as an algorithm in the cluster analysis, as it is commonly used to classify individuals based on the extraction of patterns from a dataset of 25 food group categories.

The chi-squared test was used to compare categorical variables such as age, sex, household income, province, and physical activity. The descriptive comparisons were presented as least squares means ± standard error (SE), adjusted for age, sex, BMI, and energy intake (kcal), MET score (METs), and house income using analysis of covariance (ANCOVA). Continuous variables, such as BMI, HGS, HGSWR, BP, and nutrients, and food groups were adjusted for age and sex and tested with the Bonferroni post-hoc multiple comparison after one-way analysis of variance (ANOVA) and ANCOVA using the SURVEYREG procedures. Finally, multiple regression analysis was used to evaluate HGS as a function of the food components.

All analyses were performed using SAS version 9.4 (SAS version 9.4, SAS Institute, Inc., Cary, NC, USA). Statistical significance was defined as a *p* value < 0.05.

## 3. Results

Most of the participants were categorized into the *ready-to-eat* group. About 85.6% of the subjects were attributed to a *ready-to-eat* dietary pattern, 9.3% of the subjects to a *Western-style fast-food* dietary pattern, and 5.1% participants to a *balanced* dietary pattern.

Table 1 shows the general characteristics of the study participants as separated by dietary patterns. In the *Western-style fast-food* group, 51.1% of the participants were 16–18 years old, while 18.4% were 10–12 years old (*p* < 0.001). The *Western-style fast-food* group had the highest proportion of boys (68.6%, *p* < 0.001). With regards to anthropometric variables, adjusted BMI (20.8 ± 0.3 *p* < 0.001), BP (systolic: 110.8 ± 0.8 mmHg, diastolic: 68.0 ± 0.7 mmHg, *p* < 0.001), and adjusted BP (systolic: 109.0 ± 0.8, diastolic: 66.6 ± 0.7, *p* < 0.001) were the highest in the *Western-style fast-food* group. Gender difference in HGS was observed (boys: 30.9 ± 0.4 vs. girls: 21.9 ± 0.2, *p* < 0.001), with boys tending to have higher HGS values than girls (Appendix A). HGS and HGSWR were higher in participants with a *balanced* dietary pattern than in participants with a *ready-to-eat* pattern group following age-, sex-, and total energy intake-based adjustments (26.6 ± 0.7 and 50.0 ± 1.0, *p* < 0.001). 

The food and nutrient intake rates of the study participants are summarized in Table 2. Significant differences in total energy (kcal/day), protein (g/day), fat (g/day), and carbohydrate (g/day) intake according to the three dietary patterns were detected (*p* < 0.05).

More specifically, higher total energy, protein, and fat consumption rates were shown in the *Western-style fast-food* group. Among the food groups, higher consumption rates of white rice, wheat/bread, eggs, fish, and dairy products were shown in the *ready-to-eat* group (*p* < 0.05), whereas in the *balanced* group, higher consumption rates of whole grains, vegetables, fruits, seaweed, and dairy products were found (*p* < 0.05). By contrast, a higher intake of wheat/bread, meat, meat products, oils, and soda was reported in the *Western-style fast-food* (*p* < 0.05).

A multivariable adjusted regression analysis was performed to evaluate the associations between dietary patterns and HGS. The results are presented as adjusted coefficients and 95% confidence intervals for the *ready-to-eat* and *Western-style fast-food* patterns, which were compared with the *balanced* pattern (Table 3).

Compared with the *balanced* dietary pattern, the *ready-to-eat* and *Western-style fast-food* patterns were shown to have lower nutrient content. Notably, higher intakes of wheat/bread, meat (including its derivatives), and oils, as well as lower intakes of fruit, were shown in the *ready-to-eat* and *Western-style fast-food* patterns; in addition, these two last patterns were also shown to imply a lower intake of carbohydrates, fibers, calcium, and vitamin C than the *balanced* diet pattern.

Multivariate linear regression analysis was used to evaluate the association between dietary patterns and HGS and HGSWR in the 10–18-year-old children and adolescents, as presented in Table 4. Both Model 1 (i.e., adjustment for age, sex, and house income) and Model 2 (i.e., adjustment for age, sex, BMI, energy intake (kcal), METs, and house income) were inversely associated with HGS and HGSWR (*p* < 0.05). Compared with the *balanced* pattern, the *ready-to-eat* pattern was shown to be associated with significantly lower HGS and HGSWR. However, no significant association was found between HGS or HGSWR in the *Western-style fast-food* pattern compared to *balanced* pattern.

## 4. Discussion

According to our results, there was a higher proportion of *ready-to-eat* dietary patterns than we expected in Korean children and adolescents, particularly *ready-to-eat* dietary patterns with high consumption of white rice, wheat/bread, eggs, fish, and dairy products. The *ready-to-eat* dietary pattern seems to have unhealthy effects on children and adolescents due to its hypocaloric and nutrient-poor content. Moreover, HGS was lower in participants from the *ready-to-eat* group than those from the *balanced* and *Western-style fast-food* groups. Further, following adjustment for potential confounders, HGSWR was higher in participants following a *balanced* or *Western-style fast-food* dietary pattern rather than in those following a *ready-to-eat* dietary pattern. In a study of in 1086 adolescents, McNaughton et al. showed that a dietary pattern rich in fruit, salad, cereals, and fish might be associated with lower diastolic BP in adolescents [25]. Moreover, recent studies showed that greater HGS is associated with longitudinal health maintenance and health improvements in adolescents [26]; in addition, Ng et al. reported that low HGS could serve as a prognostic indicator of cardiometabolic risk and could help to identify adolescents who would benefit most from lifestyle interventions to improve muscular fitness [26]. Remarkably, our results agree with those of the aforementioned studies. However, in the present study, the additional evaluation of HGSWR deepened the investigation of relative muscle strength; HGSWR seems to also be associated with the dietary intake of adolescents, especially with the *ready-to-eat* dietary pattern.

Furthermore, the participants following a *ready-to-eat* or *Western-style fast-food* dietary pattern showed a lower consumption of carbohydrates, fibers, calcium, and vitamin C compared to those following a *balanced* dietary pattern. The *ready-to-eat* dietary pattern implies a high intake of eggs, fish, processed products, and refined carbohydrates—such as white rice, flour, and bread—as well as a low intake of vegetables and fruits, which is rather common for foods from convenience stores or street-food stalls. By contrast, the *balanced* dietary pattern is crucial for adolescents, especially when active physical and emotional development is considered. Nevertheless, Korean adolescents seem to prefer simple meals such as bread, sandwiches, and cereals, and the frequency of convenient dietary patterns is increasing [27]. In addition, the data from the Korean Student Health Examination and the Korean Youth Risk Behavior Web-based Survey showed that the consumption of healthy foods such as fruits and vegetables tended to decrease, while the consumption of foods such as fast food and ramen increased [28,29]. According to the Korean Youth Risk Behavior Web-based Survey (*n* = 62,276), 39.3% of the Korean adolescents consumed convenient food more than once or twice a week, whereas 26.0% more consumed it more than thrice a week. High school students had a higher intake of convenient food than middle school students. Adolescents who consumed convenient food more than thrice a week had a lower intake of recommended foods such as fruits, vegetables, and milk, as well as higher intakes of fast food, snacks, and soda, than the rest of the adolescent population. However, this is not only a problem of Korean adolescents; in fact, the food consumption patterns of Indian adolescents have also been shown to imply inadequate dietary intakes, including lower consumption of vegetables and higher consumption of energy-dense snacks [30]. Cutler et al. reported increases in fast-food consumption patterns in older boys and girls [31]. In the United States, fast food is consumed by one-third of children each day and by two-thirds of children every week [32]. In summary, adolescents’ preference for *ready-to-eat* dietary patterns seems to be a problem worldwide [33].

The participants following a *balanced* dietary pattern ate a lot of whole grains, potatoes, beans, nuts, vegetables, mushrooms, fruits, eggs, fish, seaweed, and dairy products, as the name *balanced* suggests; notably, their HGSWR was significantly higher than that of the participants following a *ready-to-eat* dietary pattern. *Western-style fast-food* dietary patterns, which were based on wheat, bread, noodles, sugars, meats, oils, sodas, and seasonings, were not associated with decreased HGSWR compared to *balanced* dietary patterns in this study. This may be due to the inclusion of unique Korean eating patterns in the definition of the *Western-style fast-food* pattern group. Remarkably, kimchi—a traditional Korean food—was consumed in all groups; this may deviate from the traditional *Western-style diet* as we know it. In this study, we defined the *Western-style fast-food* dietary pattern as based on wheat/bread, noodles, sugar, meat, oils, sodas, and seasonings. However, other studies defined the *Western-style diet* as based on poultry, eggs, mayonnaise, fast foods, pizza, pies, and fried potatoes [33]. As a consequence, further studies are still necessary in order to find an association between the *Western-style diet* and HGSWR.

Interestingly, high BMI and low socioeconomic status are known to be associated with decreased muscle strength [34,35]. However, in this study, a reduction of HGSWR was observed in association with the *ready-to-eat* group rather than with the *balanced* group, even with adjustments for BMI and socioeconomic status. Therefore, the results of this study show that dietary patterns could independently play an essential role in relative muscle strength.

Robinson et al. showed an association between nutrition and sarcopenia and insisted that efforts to support the development of muscle strength should be started earlier in life with the optimization of diet and nutrition [36]. Moreover, unhealthy diets are well-known risk factors for several pathologies [37]. Fiber intake, fat, and physical activity have been introduced as distal risk factors in a chronic disease prevention model, and specific strategies are underway by several countries for the prevention of chronic diseases. The participants with the *ready-to-eat* dietary pattern had low caloric intakes and were more likely to ingest nutrient-poor content, with lower consumption of carbohydrates, fiber, calcium, and vitamin C when compared with the *balanced* diet. Children and adolescents need good quality food and nutrition to thrive. The results of this study suggest that when children and adolescent intake nutrient-poor and hypocaloric food, it might cause a decrease in their HGS and HGSWR. Reduction in HGS and HGSWR by participants following a *ready-to-eat* dietary pattern indicates that Korean adolescents are also experiencing problems due to nutritional imbalances. Further, adolescents need to learn the importance of following a *balanced* diet and reducing the intake of fast food or *ready-to-eat* foods in early life. In addition, an appropriate environment is required for the spread of balanced dietary customs. National efforts are needed to create and promote nutritional policies for reducing *ready-to-eat* dietary customs and the overall environment related to nutritional imbalances.

This study has several limitations. Firstly, data stemming from a single 24-h dietary recall might not be representative of the usual intake of the children and adolescents; nevertheless, this dietary recall method has been previously used and validated in various population-based nutritional studies. Secondly, this was a cross-sectional study; and a longitudinal study might instead be a better method for detecting an association between food intake and HGSWR. Finally, MRI is still the gold standard method of measuring muscle mass; however, due to the high costs associated with MRI, it is not appropriate for use in a large-scaled population-based study. Additionally, more precise data would be possible if knee extensor muscle measurements were included instead of measures of handgrip strength. Despite these limitations, this is the first study to investigate the association between HGS, HGSWR, and dietary intake in adolescents.

## 5. Conclusions

In this study, we compared three types of dietary intake patterns with HGS and HGSWR. The results of the study showed an association between HGS and HGSWR and the type of dietary pattern followed by children and adolescents.

HGS and HGSWR were significantly reduced in children and adolescents with *ready-to-eat* dietary patterns. The lower HGS and HGSWR values do not just indicate reduced muscle strength, but also imply the possible occurrence of several concurrent problems related to reduced relative muscle strength, which might be reflected by HGS and HGSWR. The overall environmental factors potentially inducing such unhealthy dietary preferences should be investigated, and appropriate lifestyle changes in Korean adolescents should be encouraged.

## Figures and Tables

**Figure 1 nutrients-12-03048-f001:**
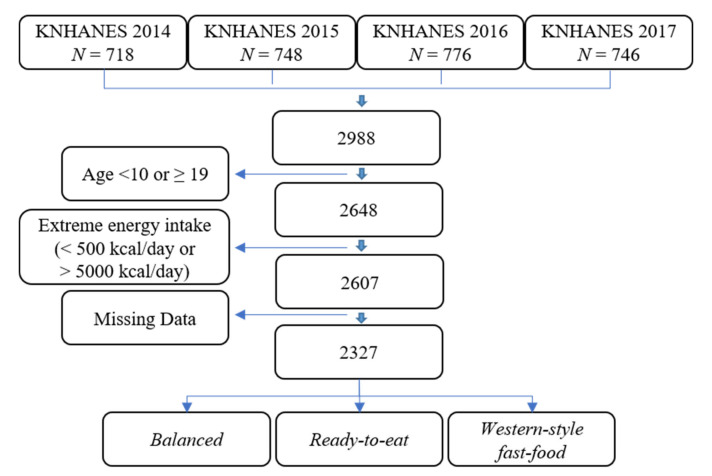
Flow chart for participant selection (2327 participants). KNHANES: Korean National Health and Nutrition Examination Survey.

**Table 1 nutrients-12-03048-t001:** General characteristics of 10–18-year-old children and adolescents according to the three dietary patterns ^1^.

	*Balanced*	*Ready-to-Eat*	*Western-Style Fast-Food*	*p*
	(*n* = 118; 5.1%)	(*n* = 1992; 85.6%)	(*n* = 217; 9.3%)	
Sociodemographics ^2^				
Age				<0.001 *
10–12 years	30.6 (4.4) 36.5 (4.7) 32.8 (5.0)	31.6 (1.2) 32.6 (1.4) 35.8 (1.3)	18.4 (2.8) 30.5 (3.4) 51.1 (3.8)	
13–15 years				
16–18 years				
Sex				<0.001 *
Boys	51.9 (4.9) 48.9 (4.9)	50.6 (1.3) 49.4 (1.3)	68.6 (3.5) 31.4 (3.5)	
Girls				
Household income				0.099
Low (Q1)	10.0 (3.3) 32.7 (5.3) 22.6 (4.3) 34.7 (5.2)	11.5 (2.4) 23.3 (1.3) 34.2 (1.5) 31.1 (1.6)	11.5 (2.4) 29.4 (3.5) 34.2 (1.5) 31.1 (1.7)	
Low-middle (Q2)				
Middle-high (Q3)				
High (Q4)				
Region				0.193
Urban	62.8 (5.1) 37.2 (5.1)	60.1 (1.7) 39.9 (1.7)	53.5 (4.0) 46.5 (4.0)	
Rural				
Physical activity				0.216
Inactive	4.7 (0.5) 5.3 (1.2) 5.5 (1.2)	85.8 (0.9) 83.0 (2.2) 78.2 (4.5)	9.5 (0.8) 11.8 (1.9) 16.4 (4.2)	
Active				
Health-enhancing				
Anthropometrics ^3^				
BMI	20.6 ± 0.4 (19.8–21.4)	20.8 ± 0.1 (20.6–21.0)	20.6 ± 0.4(20.9–22.1)	0.06
BMI *adjusted*	20.4 ± 0.4 (19.7–21.2)	20.6 ± 0.1 (20.4–20.8)	20.8 ± 0.3 (20.3–21.4)	<0.001 ^††^
BMI < 95th %tile ^2^	86.6 (3.9) 13.4 (3.9)	85.8 (0.8) 14.2 (0.8)	82.3 (2.7) 17.7 (0.8)	0.40
95^th^ %tile ≤ BMI ^2^				
HGS				
HGS	27.6 ± 1.1 (25.4–29.7)	26.1 ± 0.3 (25.5–26.6)	30.1 ± 0.8 (29.1–32.1)	<0.001 ^†^
HGS *adjusted*	26.6 ± 0.7 (25.3–27.9)	25.3 ± 0.2 (25.0–25.7)	25.8 ± 0.4 (24.9–26.6)	<0.001 ^††^
HGSWR	50.7 ± 1.3 (48.2–53.3)	47.8 ± 0.3 (47.1–48.4)	51.8 ± 1.0 (49.8–53.8)	<0.001 ^†^
HGSWR *adjusted*	50.0 ± 1.0 (47.9–52.0) ^a^	47.3 ± 0.3 (46.8–47.9) ^b^	47.7 ± 0.8 (46.2–49.2) ^ab^	<0.001 ^††^
BP				
SBP (mmHg)	108.0 ± 1.0 (105.9–109.7)	108.1 ± 0.3 (107.5–108.6)	110.8 ± 0.8 (109.3–112.3)	<0.001 ^†^
SBP *adjusted* (mmHg)	107.8 ± 1.0 (106.2–109.9) 66.2 ± 1.0 (64.3–68.1)	107.9 ± 0.3 (107.4–108.5)	109.0 ± 0.8 (107.6–110.5)	<0.001 ^††^
DBP (mmHg)		66.2 ± 0.2 (65.7–66.6)	68.0 ± 0.7 (66.0–69.3)	<0.001 ^†^
DBP *adjusted* (mmHg)	65.9 ± 0.9 (64.1–67.7)	65.6 ± 0.2 (65.2–66.1)	66.6 ± 0.7 (65.2–67.9)	<0.001 ^††^

^1^ Dietary pattern were determined by cluster analysis using dietary intake (g/day) consumed; ^2^ Values are presented as % (SE). Data were weighted to represent children and adolescents aged 10–18 years from the 2014–2017 Korean National Health and Nutrition Examination Survey (KNHANES); ^3^ Values are presented as mean ± *SE* (95% CI); * *p*-values for differences using the chi-squared test for proportions and analysis of variance for means; ^†^
*p*-values for differences using ANCOVA and adjusting for age, sex, and total energy intake (kcal); ^a, b, ab, ††^ Different superscript letters represent the results of the post hoc tests. Adjustment for multiple comparisons: Bonferroni. BMI: body mass index; HGS: handgrip strength; HGSWR: handgrip-strength-to-weight ratio; BP: blood pressure; SBP: systolic blood pressure; DBP: diastolic blood pressure.

**Table 2 nutrients-12-03048-t002:** Nutrients and food intake of 10–18-year-old children and adolescents according to the three dietary patterns.

	*Balanced*	*Ready-to-Eat*	*Western-Style Fast-Food*	
	(*n* = 118; 5.1%)	(*n* = 1992; 85.6%)	(*n* = 217; 9.3%)	*p*
Nutrients				
Total energy (kcal/day)	2418.5 ± 83.4 ^a^	2054.9 ± 19.6 ^b^	2625.2 ± 72.6 ^a^	<0.05 ^†^
Protein (g/day)	67.5 ± 2.4 ^b^	76.7 ± 0.6 ^a^	80.3 ± 2.8 ^a^	<0.001 ^††^
Fat (g/day)	45.8 ± 2.2 ^c^	59.7 ± 0.5 ^b^	64.3 ± 1.8 ^a^	<0.05 ^††^
Carbohydrate (g/day)	367.9 ± 7.4 ^a^	314.9 ± 1.6 ^b^	300.4 ± 5.5 ^c^	<0.001 ^††^
Food groups (g/day)				
White rice	137.8 ± 9.7 ^b^	165.0 ± 2.8 ^a^	106.5 ± 7.3 ^c^	<0.05 ^††^
Whole grains	35.3 ± 5.0 ^a^	26.7 ± 1.3 ^ab^	19.7 ± 3.7 ^b^	<0.05 ^††^
Wheat/bread	66.5 ± 12.5 ^b^	92.9 ± 2.8 ^a^	104.7 ± 8.5 ^a^	<0.05 ^††^
Noodle	47.5 ± 9.8	58.1 ± 2.4	41.2 ± 7.1	
Potatoes	44.8 ± 14.4	28.7 ± 1.6	30.4 ± 4.9	
Sugars	12.1 ± 2.2	15.4 ± 0.8	14.8 ± 2.1	
Beans	23.1 ± 6.0	19.8 ± 1.3	16.6 ± 3.8	
Nuts	5.8 ± 2.5	2.5 ± 0.3	2.2 ± 0.5	
Vegetables	138.2 ± 20.5 ^a^	132.7 ± 3.5 ^ab^	99.1 ± 8.9 ^b^	<0.01 ^††^
Kimchi	56.4 ± 6.5	62.2 ± 2.0	56.8 ± 7.9	
Mushrooms	8.8 ± 3.8	5.4 ± 0.5	4.1 ± 0.9	
Fruits	865.5 ± 35.5 ^a^	103.2 ± 3.6 ^b^	66.4 ± 11.2 ^c^	<0.01 ^††^
Meat and its products	77.9 ± 11.4 ^c^	134.7 ± 3.2 ^b^	199.2 ± 13.8 ^a^	<0.001 ^††^
Eggs	31.5 ± 4.9 ^ab^	32.4 ± 1.2 ^a^	20.2 ± 2.7 ^b^	<0.001 ^††^
Fish	33.7 ± 5.2 ^ab^	38.6 ± 1.9 ^a^	23.0 ± 5.1 ^b^	<0.05 ^††^
Seaweed	3.2 ± 0.9 ^a^	2.3 ± 0.2 ^a^	0.7 ± 0.2 ^b^	<0.01 ^††^
Dairy products	162.6 ± 18.7 ^a^	194.4 ± 5.6 ^a^	78.2 ± 10.1 ^b^	<0.001 ^††^
Oils	5.8 ± 0.8 ^b^	8.7 ± 0.2 ^b^	10.7 ± 1.3 ^a^	<0.01 ^††^
Soda	55.1 ± 16.4 ^b^	46.8 ± 2.7 ^b^	614.7 ± 17.8 ^a^	<0.001 ^††^
SSBs	69.8 ± 21.8	81.7 ± 4.5	78.6 ± 15.7	
Processed foods	29.9 ± 3.7	31.9 ± 0.8	32.8 ± 2.5	

Values are presented as mean ± SE; ^†^
*p*-values for differences using ANCOVA and adjusting for age, sex, and total energy intake (kcal); ^a, b, c, ab, ††^ Different superscript letters represent the results of the post hoc tests. Adjustment for multiple comparisons: Bonferroni.

**Table 3 nutrients-12-03048-t003:** Differences in food intake of 10–18-year-old children and adolescents according to the three dietary patterns.

	*Ready-to-eat*			*p*	*Western-Style Fast-Food*			*p*
Variables	1992 (85.6)				217 (9.3)			
	*Beta*	*SE*	CI		*Beta*	*SE*	CI	
Food groups (g/day)								
White rice	89.0	33.9	(22.5, 155.4)	<0.001	–118.3	44.2	(–205.0, –31.5)	<0.001
Whole grains	–22.1	13.9	(–49.5, 5.2)		–47.9	16.5	(–80.2, –15.6)	<0.05
Wheat/bread	87.7	32.6	(23.6, 151.8)	<0.01	128.4	41.0	(47.7, 209.0)	<0.01
Noodle	18.5	30.7	(–41.7, 78.7)		–26.0	38.0	(–100.6, 48.7)	
Potatoes	–10.0	12.0	(–33.7, 13.6)		12.6	14.5	(–15.9, 41.0)	
Sugars	13.7	8.5	(–3.0, 30.3)		15.7	11.6	(–7.0, 38.5)	
Beans	–5.1	6.1	(–17.1, 6.9)		–9.5	7.2	(–23.7, 4.7)	
Nuts	–6.5	4.4	(–15.1, 2.1)		–5.3	5.1	(–15.3, 4.7)	
Vegetables	–3.1	6.7	(–16.3, 10.1)		–11.6	7.5	(–26.4, 3.1)	
Kimchi	0.7	2.0	(3.2, 4.5)		–1.1	2.6	(–6.2, 4.1)	
Mushrooms	–1.4	1.4	(–4.1, 1.3)		–2.1	1.5	(–5.1, 0.9)	
Fruits	–326.7	15.9	(–357.9, –259.5)	<0.000	–344.4	17.1	(–377.9, –310.8)	<0.000
Meat and its products	113.0	27.5	(59.1, 166.9)	<0.001	244.2	41.3	(163.1, 325.4)	<0.001
Eggs	0.4	8.0	(–15.3. 16.2)		–17.8	9.1	(–35.7, 0.1)	
Fish	2.3	7.3	(–12.0, 16.6)		–15.2	9.5	(–33.8, 3.4)	
Seaweed	0.1	0.9	(–1.7, 2.0)		–2.3	1.0	(–4.4, –0.3)	<0.05
Dairy products	25.2	19.9	(–14.0, 64.4)		–67.4	22.3	(–111.3, –23.6)	<0.05
Oils	25.0	7.0	(11.1, 38.8)	<0.001	42.8	13.3	(16.6, 69.0)	<0.01
Soda	–2.2	6.9	(–15.6, 11.3)		236.1	11.0	(214.4, 257.8)	<0.001
SSBs	1.3	13.9	(–26.1, 28.7)		–5.3	15.6	(–36.0, 25.4)	
Processed foods	3.1	8.8	(–14.3, 20.4)		11.7	10.5	(–9.0, 32.4)	
Nutrients								
Total energy (kcal/day)	–369.0	82.1	(–530.3, –207.8)	<0.001	201.4	106.7	(–8.2, 411.0)	
Protein (g/day)	–4.6	3.2	(–11.0, 1.7)		20.2	5.4	(9.7, 30.8)	<0.001
Fat (g/day)	1.3	2.6	(–3.8, 6.4)		25.0	4.1	(17.0, 33.1)	<0.001
Carbohydrate (g/day)	–100.6	15.4	(–130.9, –70.2)	<0.001	–40.3	18.2	(–75.9, –4.6)	<0.05
Fiber (g/day)	–13.6	1.8	(–17.2, –10.0)	<0.001	–12.0	2.0	(–15.9, –8.1)	<0.001
Calcium (mg/day)	–70.8	31.1	(–131.8, –9.7)	<0.05	–118.0	36.4	(–189.5, –46.5)	<0.01
Vitamin A (ugRE/day)	–112.1	111.1	(–330.4, 106.1)		–118.6	122.4	(–358.9, 121.8)	
Vitamin C (mg/day)	–145.4	21.7	(–187.9, –102.9)	<0.001	–142.5	22.5	(–186.7, –98.3)	<0.001

Statistical analysis was performed using multivariate linear regression. Variables: age, sex, BMI, energy intake (kcal), metabolic equivalent task score (METs), and household income.

**Table 4 nutrients-12-03048-t004:** Multivariate linear regression analysis of the association between dietary patterns and HGS of 10–18-year-old children and adolescents.

Variables	*Ready-to-Eat*			*p*	*Western-Style Fast-Food*			*p*
	1992 (85.6)				217 (9.3)			
	*Beta*	*SE*	CI		*Beta*	*SE*	CI	
HGS								
Model 1	–1.6	0.7	(–2.9, –0.3)	0.0148	–0.5	0.8	(–2.1, 1.1)	0.5213
Model 2	–1.3	0.6	(–2.5, –0.0)	0.0423	–0.9	0.7	(–2.3, 0.6)	0.2431
HGSWR								
Model 1	–3.0	1.0	(–5.1, –0.9)	0.0044	–1.9	1.3	(–4.4, 0.5)	0.1206
Model 2	–2.2	0.9	(–3.9, –0.5)	0.0101	–1.5	1.0	(–3.5, 0.4)	0.1162

HGS: handgrip strength; HGSWR: handgrip-strength-to-weight ratio; Statistical analysis was performed using multivariate linear regression; Model 1: adjustment for age, sex, and household income; Model 2: adjustment for age, sex, BMI, energy intake (kcal), METs, and house income.

## Data Availability

Data are available at https://knhanes.cdc.go.kr/.

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
