# Peer review of "Association between Dietary Patterns and Handgrip Strength: Analysis of the Korean National Health and Nutrition Examination Survey Data Between 2014 and 2017"

_nutrients, 2020, doi:10.3390/nu12103048_

Round 1

Reviewer 1 Report

The study in well conducted and well referred.  However in your paper you do not mention the difference that gender may play in handgrip evaluation. A recent italian study in 9-10 year aged children found that gender  correlated to handgrip force (p<0.04).<< T. Montalcini et al. Gender difference in handgrip strength of Italian children aged 9 to 10 years. It J Ped 2016;42>> I think that this difference might be considered in your work as the patients are adolescents, just an age where gender difference  becomes more important for the factors you have studied.

Author Response

Thank you for your comments concerning our manuscript. Those comments were all valuable and very helpful for our paper. We have carefully read the comments and made correction which we hope to meet with approval. Revised responses to editor and reviewer are marked in the paper using ‘track changes’ function.

Reviewer 1:

The study in well conducted and well referred. However in your paper you do not mention the difference that gender may play in handgrip evaluation. A recent italian study in 9-10 year aged children found that gender correlated to handgrip force (p<0.04).<< T. Montalcini et al. Gender difference in handgrip strength of Italian children aged 9 to 10 years. It J Ped 2016;42>> I think that this difference might be considered in your work as the patients are adolescents, just an age where gender difference becomes more important for the factors you have studied.

-> Thank you for your comment. We appreciate to your comment and consideration of the present study. As your opinion, we performed statistical analysis before we started analyzing the association between dietary patterns and HGS in children and adolescents (supplement 1). Consistent with the previous study (T. Montalcini et al.), we found gender difference in HGS (Boys: 30.9 ± 0.4 vs. Girls: 21.9 ± 0.2, p <.0001).

However, we aimed to investigate the association between and dietary patterns and handgrip strength (HGS) in Korean adolescents. Therefore, we more focused to describe and analyze the differences of each dietary patterns; balanced, ready-to-eat and western-style fast-food of the participants and HGS. Following as our study aims, we considered that not only gender differences also age, household income, region, PA and BMI (adjusted) among children and adolescents in Korea according to the three dietary patterns (Table 1).

We added supplement data 1 about gender differences and add sentence as following

Line 165-166: Gender difference in HGS is seen (Boys: 30.9 ± 0.4 vs. Girls: 21.9 ± 0.2, p <.0001) and boys tend to have higher HGS than girls. (Suppl 1.)

Supplementary Data 1. General difference according to gender and age

Gender

Age

All

Boys

Girls

10~12yr

13~15yr

16~18yr

N

2,327

1,223

1,104

865

807

655

Age

14.2 ± 0.11)

14.2 ± 0.1

14.2 ± 0.1

11.0 ± 0.0

14.1 ± 0.0

16.9 ± 0.0***

BMI

20.9 ± 0.1

21.1 ± 0.1**

20.6 ± 0.1

19.2 ± 0.1

21.1 ± 0.1

22.1 ± 0.2***

95th%tile≤ BMI

329(14.1%)

200(16.3%)*

129(12.6%)

77(9.1%)

119(13.7%)

133(19.6%)***

HGS

26.6 ± 0.3

30.9 ± 0.4***

21.9 ± 0.2

18.2 ± 0.2

28.3 ± 0.3

32.0 ± 0.4***

SBP 2)

108.4 ± 0.3

109.5 ± 0.34***

106.5 ± 0.4

106.3 ± 0.4c

108.5 ± 0.4b

109.9 ± 0.4a

DBP

66.4 ± 0.2

66.8 ± 0.3***

65.9 ± 0.3

62.8 ± 0.3c

66.0 ± 0.3b

69.6 ± 0.4a

Reviewer 2 Report

The authors present an interesting article investigating the relationship between handgrip strength and dietary pattern in an adolescent population. The authors used a large sample of adolescents from ages 10 18 years old taken from the Korean National Health and Nutrition Examination Survey.

Specific comments

Line 45-46: There are additional methods that can quantify skeletal muscle mass such as the gold standard using MRI and ultrasound. Therefore can the authors add this to the introduction.

Line 46-47: The authors specify movements that strength can be assessed in and not the method used to assess muscle strength. Can the sentence please be amended to reflect this.

Line 49-50: Muscle quality is measured by dividing muscle strength by muscle muss of the particular limb, where I believe they are referring to relative strength. Therefore, can the authors please amend this statement

Line 219: Handgrip strength to body weight ratio does not indicate muscle quality by more relative strength. Therefore, can this statement please be amended to reflect this statement.

Line 262: See comments above about the authors’ use of muscle quality within the manuscript instead of relative strength.

Line 279 -285 Can the authors expand upon the limitations in more detail. One important limitation is not defining muscle quality using one of the methods described within the introduction. MRI would be the gold standard but ultrasound and dual energy x-ray absorptiometry would be applicable for the specific muscle group utilised. Another limitation is the choice of muscle group/action utilised because a loaded muscle group such as the knee extensors would be a more precise indicator of functional ability, followed with the inclusion of power-based tests. Finally, do the authors believe 24-hour recall or interviews provide a true representation of an individual’s dietary intake over a period of one month or longer? Therefore, can the authors provide a reference to support this statement or add to the limitations section.

Line 287 – 295: can the authors use relative strength instead of muscle quality because that has not been measured

Author Response

Thank you for your comments concerning our manuscript. Those comments were all valuable and very helpful for our paper. We have carefully read the comments and made correction which we hope to meet with approval. Revised responses to editor and reviewer are marked in the paper using ‘track changes’ function.

Reviewer 2:

Line 45-46: There are additional methods that can quantify skeletal muscle mass such as the gold standard using MRI and ultrasound. Therefore can the authors add this to the introduction.

-> Thank you for your comment. I added the sentences as follow

Skeletal-muscle mass can be assessed by either dual-energy X-ray absorptiometry or bioelectrical impedance analysis.

-> Skeletal-muscle mass can be assessed by either dual-energy X-ray absorptiometry or bioelectrical impedance analysis and magnetic resonance imaging, ultrasound can be used to measure skeletal muscle quantity

Line 46-47: The authors specify movements that strength can be assessed in and not the method used to assess muscle strength. Can the sentence please be amended to reflect this.

-> Thank you for your comment. I rephrased the sentence.

Furthermore, muscle strength can be measured using, plank exercise, modified pull-up, knee extension and handgrip strength

-> Furthermore, muscle strength can be accessed by plank exercise, modified pull-up, knee extension and handgrip strength

Line 49-50: Muscle quality is measured by dividing muscle strength by muscle muss of the particular limb, where I believe they are referring to relative strength. Therefore, can the authors please amend this statement

-> Thank you for your comments. I amended the sentence as follow:

Furthermore, muscle quality can be estimated when handgrip strength is divided by body weight.

-> Furthermore, muscle quality can be estimated by dividing muscle strength by muscle mass size. However, in a simple way, we can try to predict relative muscle strength by dividing handgrip strength by body weight.

Line 219: Handgrip strength to body weight ratio does not indicate muscle quality by more relative strength. Therefore, can this statement please be amended to reflect this statement.

-> Thank you for your comment. I rephrased the sentence as follow

However, in the present study, the additional evaluation of HGSWR deepened the investigation of muscle quality; besides, HGSWR seems to be also associated with dietary intake of adolescents, especially with the ready-to-eat dietary pattern.

-> However, in the present study, the additional evaluation of HGSWR deepened the investigation of relative muscle strength; besides, HGSWR seems to be also associated with dietary intake of adolescents, especially with the ready-to-eat dietary pattern.

Line 262: See comments above about the authors’ use of muscle quality within the manuscript instead of relative strength.

-> Thank you for your comment. I rephrased the sentence with ‘muscle quality’.

Line 30

association between reduced muscle quality and dietary pattern.

-> association between reduced relative muscle strength and dietary pattern.

Line 264

Therefore, the results of this study show that dietary patterns could independently play an essential role in muscle quality.

-> Therefore, the results of this study show that dietary patterns could independently play an essential role in relative muscle strength.

Line 293-294

HGS and HGSWR in ready-to-eat dietary patterns were significantly reduced, and it does not just indicate reduced the quality of simple muscles. This implies the possible occurrence of several concurrent problems related to reduced muscle quality, which might be reflected by HGS and HGSWR.

-> HGS and HGSWR in ready-to-eat dietary patterns were significantly reduced, and it does not just indicate reduced the relative muscle strength. This implies the possible occurrence of several concurrent problems related to reduced muscle strength, which might be reflected by HGS and HGSWR.

Line 279 -285 Can the authors expand upon the limitations in more detail. One important limitation is not defining muscle quality using one of the methods described within the introduction. MRI would be the gold standard but ultrasound and dual energy x-ray absorptiometry would be applicable for the specific muscle group utilised. Another limitation is the choice of muscle group/action utilised because a loaded muscle group such as the knee extensors would be a more precise indicator of functional ability, followed with the inclusion of power-based tests. Finally, do the authors believe 24-hour recall or interviews provide a true representation of an individual’s dietary intake over a period of one month or longer? Therefore, can the authors provide a reference to support this statement or add to the limitations section.

Thank you for your comment. We deeply understand what your opinion about the methodological limitation of the dietary assessment. We added the in detail limitation in the revised manuscript as below.

Nonetheless, this study has several limitations. Firstly, errors might occur in the 24-h dietary recall method in face-to-face interviews even when examinations are performed by trained interviewers. Secondly, many common types of food might not be represented in the questionnaire. Thirdly, this is a cross-sectional study, and a longitudinal study might instead be a better method for detecting an association between food intake and HGSWR.

-> Nonetheless, this study has several limitations. Firstly, the single 24-h dietary recall method might not be represented usual intake of the children and adolescents nevertheless, it has been validated and developed in various population based nutritional studies. Secondly, this is a cross-sectional study, and a longitudinal study might instead be a better method for detecting an association between food intake and HGSWR. Finally, MRI is still gold standard method of measuring muscle mass, however, due to their high cost, it is not appropriate to use in large scaled population-based study. Additionally, more precise data would be possible if knee extensor muscle measurement were included instead of handgrip strength.

Line 287 – 295: can the authors use relative strength instead of muscle quality because that has not been measured

-> Thank you for your comment. I rephrased the muscle quality to ‘relative muscle strength’

HGS and HGSWR in ready-to-eat dietary patterns were significantly reduced, and it does not just indicate reduced the quality of simple muscles. This implies the possible occurrence of several concurrent problems related to reduced muscle quality, which might be reflected by HGS and HGSWR.

-> HGS and HGSWR in ready-to-eat dietary patterns were significantly reduced, and it does not just indicate reduced simple muscles strength. This implies the possible occurrence of several concurrent problems related to reduced relative muscle strength, which might be reflected by HGS and HGSWR.

Round 2

Reviewer 2 Report

The authors have now addressed all of my concerns